# A modular organic neuromorphic spiking circuit for retina-inspired sensory coding and neurotransmitter-mediated neural pathways

Giovanni Maria Matrone [1,2,9] ✉, Eveline R. W. van Doremaele [1,9], Abhijith Surendran [2], Zachary Laswick[2], Sophie Griggs[3], Gang Ye[4], Iain McCulloch [3,5], Francesca Santoro[6,7,8], Jonathan Rivnay [2] & Yoeri van de Burgt [1] ✉

Signal communication mechanisms within the human body rely on the transmission and modulation of action potentials. Replicating the interdependent functions of receptors, neurons and synapses with organic artificial neurons and biohybrid synapses is an essential first step towards merging neuromorphic circuits and biological systems, crucial for computing at the biological interface. However, most organic neuromorphic systems are based on simple circuits which exhibit limited adaptability to both external and internal biological cues, and are restricted to emulate only specific the functions of an individual neuron/synapse. Here, we present a modular neuromorphic system which combines organic spiking neurons and biohybrid synapses to replicate a neural pathway. The spiking neuron mimics the sensory coding function of afferent neurons from light stimuli, while the neuromodulatory activity of interneurons is emulated by neurotransmitters-mediated biohybrid synapses. Combining these functions, we create a modular connection between multiple neurons to establish a pre-processing retinal pathway primitive.

Neuromorphic computing is currently experiencing a tremendous increase in research activity. Inspired by neuroscience, it aims to mimic the functions and architecture of the brain to enable parallel computing with high energy efficiency[1,2] and local processing[3], which advances edge-computing[4], smart robotics[5,6] and intelligent systems interfacing with the human body[7,8]. However, establishing an active interaction with biological tissues, especially with the central nervous system, requires computing systems that are not only able to receive biologically encoded inputs but also to process and communicate these, by adapting their processing functions to cues such as neurotransmitters and light. The complex sensory systems in the human body share a coding mechanism based on the frequency modulation of action potentials[9]. The physical perception starts with millions of highly specific sensory receptor cells which respond to definite stimuli at distinct locations on the body. The intensity of an external stimulation is first converted into a receptor potential and then encoded in

[1]Microsystems, Institute for Complex Molecular Systems, Eindhoven University of Technology, 5612AJ Eindhoven, The Netherlands. [2]Department of Biomedical Engineering, Northwestern University, Evanston, IL 60208, USA. [3]Department of Chemistry, Chemistry Research Laboratory, University of Oxford, Oxford OX1 3TA, UK. [4]Center for Biomedical Optics and Photonics (CBOP) & College of Physics and Optoelectronic Engineering, Key Laboratory of Optoelectronic Devices and Systems, Shenzhen University, Shenzhen 518060, PR China. [5]King Abdullah University of Science and Technology (KAUST), KAUST Solar Center (KSC), Thuwal 23955-6900, Saudi Arabia. [6]Tissue Electronics, Istituto Italiano di Tecnologia, Naples 80125, Italy. [7]Institute of Biological Information Processing IBI-3 Bioelectronics, Forschungszentrum Juelich, 52428 Juelich, Germany. [8]Neuroelectronic Interfaces, Faculty of Electrical Engineering and IT, RWTH Aachen, 52074 Aachen, Germany. [9]These authors contributed equally: Giovanni Maria Matrone, Eveline R. W. van Doremaele. ✉e-mail: giovanni.matrone@northwestern.edu; Y.B.v.d.Burgt@tue.nl

the firing frequency of afferent neurons. The direct dependence of the spike frequency on the magnitude of the stimulus is generally referred to as sensory coding. The sensory information is then transmitted to the central nervous system where it is processed through a complex network of inhibitory/excitatory interneurons that work as computing units by modulating the spike frequency in parallel[10,11].

The circuital design of current spiking circuits is predominantly based on inorganic materials, particularly on silicon-based devices[12,13] (Supplementary Discussion 1, Supplementary Table 1). However, neuromorphic applications operating at the biointerface clearly favour organic materials[2,14]. Next to their relatively soft and flexible properties, high tunability and low operational voltage, organic mixed ion-electron conductors conduct both electrons and ions[15]. Although organic non-electrolyte conductors have been previously employed to build spiking circuits based on organic field effect transistors (OFETs), their lack of ionic conduction (interaction with aqueous electrolytes) severely limit their applicability towards adaptive neuromorphic circuits[16]. Standard OFETs operate in dry conditions and do not require an electrolyte medium, which in the context of neuromorphic and smart bioelectronic applications represents the bio-interfaced environment interacting with the platform[17]. Moreover, to act as biosensors these devices usually require surface functionalisation and suffer from limitations related to low sensitivity and selectivity.

Mixed ionic-electronic conductors allow to build organic electrochemical transistors (OECTs) where ion injection from an electrolyte modulates the bulk conductivity of the channel terminal. This mechanism allows to design neuronal circuits that closely match the operating timescales of their biological counterparts[18], reduce interface impedance[19], and mimic ion-based biological functions such as neuronal ion-flux communication and neurotransmitter-receptor binding[2] (synaptic function), which are required to interact with biological tissues and to design adaptive biointerfaces[20]. Recently, organic materials have been employed to build electronic circuits that mimic the spiking behaviour of neurons[16,21,22]. In this context, mixed ionic-electronic conductors-based neuromorphic devices are emerging due to their intrinsic ion tunability which elicit a change of their electrical characteristics[23]. The distinctive interdependence of these conduction mechanisms (ionic to electronic) is essential to replicate the chemical (neurotransmitters) to electrical (action potential) signal transduction processes in-sensor and is performed by the two key biological elements considered in this work: synapses and neurons[22,24,25]. On the contrary, to perform similar biological functions inorganic and OFETs-based circuits devices require a large number of circuital elements with limited tissue adaptation and complex external sensing units. These require multiple signal transmission optimisation mechanisms (amplification, reduction, noise control), significantly limiting the reconfigurability of the overall circuit[26–28].

Despite their ability to process complex sensorial input[29,30], the most recent organic systems operate as stand-alone neurons, transducing external signals such as light, pressure, temperature, ionic concentrations etc. into spike patterns, but they lack mechanisms to transmit the encoded signal to different processing units in the neural network (Supplementary Discussion 1, Supplementary Table 1). This signal transmission and modulation is the fundamental property of interneurons[31] and is essential to perform pre-processing functions and overcome the limit of conventional electronic spiking circuits[32,33]. Organic materials can be used to recreate a neuronal network that relies on the cooperation of spiking elements (afferent neurons and interneurons) and non-spiking elements such as mechano-chemical sensors (receptors), as well as neuromodulator junctions (chemical synapses). In this work, we demonstrate this concept by designing and fabricating monolithically integrated neuromorphic spiking circuits (neurons) that are connected through biohybrid (chemical) synapses to develop a modular system that emulates the biological functions of a retinal pathway.

## Results

### A neuromorphic system for sensory coding and neuromodulation

The organic neuromorphic system is able to replicate the spiking activity of both afferent neurons (sensory coding through external stimuli such as light), as well as interneurons (neuromodulation with neurotransmitters on a biohybrid synapse[24]), see Fig. 1a. The spiking neurons are fabricated on a $16 \times 16$ mm glass slide (see Fig. 1b and Supplementary Discussion 1). They consist of a pair of ambipolar inverters (p($C_4$-T2-$C_0$-EG)[34]), comprising four OECTs and a resetting electrochemical transistor (P-3O[35]). The receptor which includes a light sensor converts the external stimulus to an input potential that activates the neuromorphic spiking circuit. The input potential charges the spiking circuit (through the intrinsic capacitance of the OECTs) until the transition voltage of the inverter pair is reached. The voltage inversion triggers the resetting OECT which allows the circuit to discharge and start the next cycle. The fabrication details as well as the characterisation of both the spiking neuron and the individual devices can be found in Supplementary Discussion 1.

Figure 1c demonstrates the sensory coding of light stimuli for low and high ambient light intensity (see Supplementary Discussion 2 for sensory coding of intermediate intensity light stimuli). We connect a commercial light sensor that replicates the function of the cones in the retina (the main physical receptor in bright light conditions), transducing different light intensities to specific receptor potentials that serve as the input to the spiking neuron, which thus emulates the main mechanism of afferent neurons (sensory coding). Each input generates a train of voltage spikes with a characteristic frequency ranging from 0.09 Hz to 0.25 Hz depending on the light intensity, corresponding to a maximum 137% modulation (Supplementary Discussion 1 and 2).

Although afferent neurons are fundamental for sensory coding, they represent only a small proportion of neurons in the human body, while interneurons represent 99% of the total[9]. Besides being responsible for transmitting the signal between different areas of the brain, their crucial task is to locally modulate the spiking pattern (spike-rate coding) through different neuromodulators at chemical synapses, in a process called neuromodulation which enables the essential functions of learning and decision-making[36]. Our neuromorphic system is designed to mimic the neuromodulation process of an interneuron using the neurotransmitters dopamine (DA) and serotonin (5-HT) as biological cues. Dopamine (DA) is a well-known neuromodulator essential for motor regulation functions, rewards, and addiction[37] while serotonin is involved in controlling mood, perception, memory and attention[38]. In addition, recent studies have addressed the involvement of dopamine in information processing, neurogenesis and diurnal patterns showing how this neurotransmitter influences sensory coding systems[39]. We replicate the connection between an interneuron and a neighbouring neuron using an organic neuromorphic device within a synaptic modulator circuit (biohybrid synapse, see Fig. 1b, d). This device adopts a configuration similar to an organic electrochemical transistor (OECT), as previously reported (see Supplementary Discussion 3), and its conductance can be modulated in a volatile way upon ion injection by voltage spikes at the gate. When an electroactive neurotransmitter, such as dopamine or serotonin, is present in the electrolyte, the oxidation of these species leads to a non-volatile conductance modulation (synaptic weight modulation)[24,40]. This mechanism and the use of a fluidic module (Supplementary Discussion 3) replicate the phasic release of neurotransmitters (in the midbrain and retina), which is assumed to be associated to the biological functions of signal processing leading to learning and memory[41]. Signal computing at the synaptic level depends on the fluctuation of neurotransmitters concentration in the mM range at the synaptic cleft (the pre- to post-synaptic 4 μm space), as these are released upon presynaptic firing (Supplementary Discussion 3 and 5). Hence, as an input to the synaptic modulator, we connected a spiking neuron which

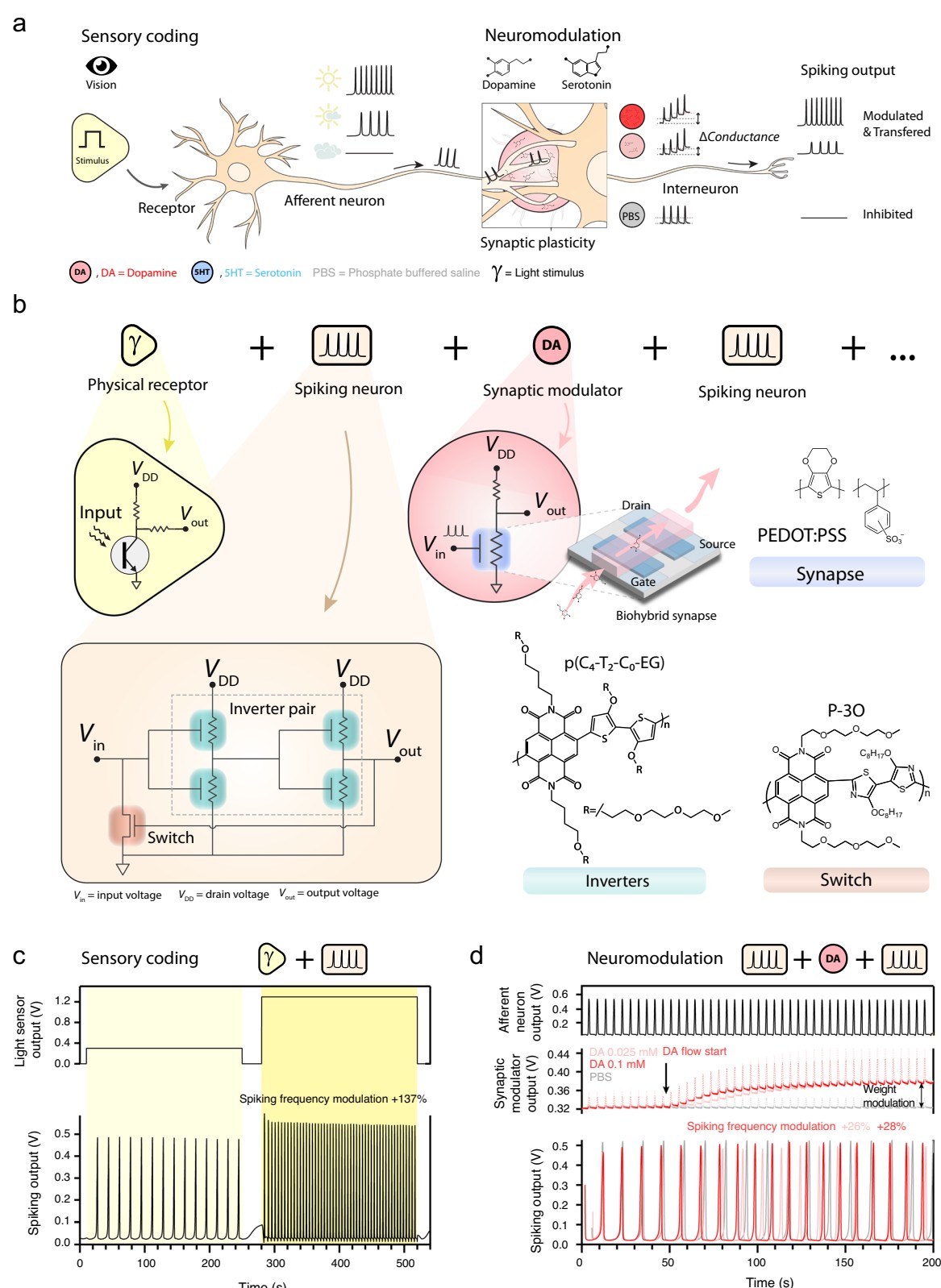

imitates the firing (i.e., spiking activity) of an afferent neuron (Fig. 1d). These spikes are applied to the gate of the OECT. Since their voltage amplitude is greater than the dopamine oxidation potential ($Vox_{DA} = 0.3\,V$), a non-volatile conductance change is induced which causes the output of the synaptic modulator to increase (Fig. 1d). Hence, the synaptic modulator represents the weight of the biohybrid synaptic connection. The next spiking neuron in our hybrid pathway,

the DA-interneuron, receives this variable output voltage which directly affects the spike frequency (Fig. 1d). Note that the output of the synaptic modulator shows spikes (dotted lines Fig. 1d) which are the result of the spiking input signal. However, only the baseline voltage (solid lines in Fig. 1d) represents the synaptic weight and has an impact on the spike frequency (output) of the connected DA neuron (Fig. 1d and Supplementary Discussion 4). Figure 1d shows the

**Fig. 1 | Neuromorphic spiking system emulating sensory coding and neuro-modulation. a** Illustration representing sensory coding of light stimuli by the afferent neuron and dopamine-regulated neuromodulation by the interneuron. **b** Schematic of the modular neural pathway including the neuromorphic spiking neuron consisting of an inverter pair and a switch with their corresponding active materials. For sensory coding a physical receptor (an ambient light sensor) is connected and for neuromodulation a synaptic modulator, comprising a biohybrid synapse using PEDOT:PSS, is employed. **c** Sensory coding of light using an ambient light sensor as physical receptor showing an increasing spike frequency of the afferent neuron output with increasing light intensity and corresponding sensor output voltage (see Supplementary Discussion 1). The output frequency shows a

137% modulation with respect to the spike frequency (0.10 Hz) of the first (low) light condition. **d** Neuromodulation regulated by dopamine using the synaptic modulator circuit. The graph shows the input to the synaptic modulator and the modulation of the interneuron spike frequency depending on the dopamine concentration. The synaptic modulator output increases due to the oxidation of DA (red). The dotted lines are the experimental curves showing the presence of the spiking input. The solid lines emphasise the synaptic weight modulation. The spike frequency modulations are 28% and 26% for 0.1 mM DA (dark red) and 0.025 mM DA (light red), respectively, with respect to the spike frequency of PBS (black; 0.10 Hz).

modulation of the interneuron spike frequency for two different concentrations of dopamine, 0.025 mM (light red) and 0.1 mM (dark red), which simulate the instantaneous concentrations of neurotransmitters at the synapse during firing, while the afferent neuron is spiking during illumination. The introduction of dopamine to the system (black arrow) leads to a gradual increase of the synaptic modulator output from 0.32 to 0.38 V, and consequently an increase in the spike frequency of the neuron (+26% and +28% for 0.025 mM and 0.1 mM DA, respectively). These modulations depend on the concentration of the neurotransmitter in the interneuron biohybrid synapse, similar to the to the biological mechanism of synaptic plasticity[42] as well as on the level of illumination (afferent neuron spike frequency) (see Supplementary Discussion 4). Due to its modular elements, specifically the spiking circuit and synaptic modulator, our neuromorphic system is able to replicate the main functions of both afferent neurons and interneurons, which are responsible for sensory coding and spike-rate coding through neuromodulation, respectively. However, in biological systems the processing of signals requires a large collective set of computing elements (neurons and synapses).

## Neural pathway between dopaminergic and serotonergic neurons

In the brain, every single neuron computes the spikes received from many interneuron connections and fires a spiking output if the sum of the input stimuli has reached a specific threshold[9]. Neural pathways, the connections formed by an axon and its synapse, link neurons from different locations and enable the transmission of signals from one region of the nervous system to another[43]. The modularity of our neuromorphic system allows to mimic the circuitry of the nervous system and design cascading circuits which closely resemble the biological connections within a neural pathway.

We replicate an interdependent neural connection between two interneurons, where a spiking signal is transferred or inhibited by the sending neuron whose spiking activity, in turn, triggers the spiking of the next interneuron (receiving neuron) in the biological circuit[44]. The first interneuron is regulated by a synapse expressing DA and computes and transmit the signal through a 5-HT synapse to the second neuron (see Fig. 2a). Dopaminergic and serotonergic neurons are often interwoven through interneuron connections even if located in different brain areas[44,45].

In this demonstration, the input signal to the neural pathway (presynaptic signal) consists of periodic voltage spikes generated by customized software to represent the spikes generated by a generic (afferent) neuron. Here we prove that the connected organic neurons and their synapses are able to replicate the biological dependency where the spiking activity of the receiving (serotonergic) neuron depends on the activity of sending (dopaminergic) neurons[46].

Essentially, the neurotransmitters act as a trigger to overcome the threshold that activates the firing characteristics of neurons, thus forming an artificial neural pathway with the connected spiking neurons and synapses.

The DA and 5-HT regulated spiking neurons are connected through their synapse with the synaptic modulators (Supplementary

Discussion 3, 4 and 5) (Fig. 1b). Both synapse-spiking circuits (DA-neuron and 5-HT-neuron) follow the same dynamics which allows the exchange of the sending and receiving neuron position in the cascade circuit, further highlighting the modularity and reconfigurability of the system. In the experiments of Fig. 2, the DA-neuron receives the presynaptic input voltage and the signal propagation along the neural pathway (visualized in Fig. 2a) is monitored in Fig. 2b. The output of the synaptic modulator and the output of the spiking circuit are shown in in Fig. 2b, of both DA (red) and 5-HT (blue) neurons.

At the start of the experiment (0–40 s), the DA-neuron does not generate any spikes and behaves as a silent neuron since the output of the synaptic modulator ($V_{Synmod} = 0.1$ V) is below the threshold voltage ($V_{threshold} = 0.2$ V, Supplementary Discussion 1, Supplementary Discussion 1, Supplementary Figs. 7 and 15). After 40 s DA is introduced to the system and the output of the synaptic modulator increases from 0.12 V to 0.2 V (see red line Fig. 2b). In this case, the neural pathway excitatory functions are enabled due to the oxidation of DA (see Supplementary Discussion 3). As a result, the synaptic modulator output increases exceeding the threshold voltage of the neuron and activates its firing (see red line of the neuron output Fig. 2b). At this point, 5-HT is absent, and the last (serotonergic) neuron of the pathway shifts from the silent to the failed activation state where only transient spikes are recorded (neuron output grey lines Fig. 2b). However, when 5-HT is introduced (blue lines Fig. 2b), the output of the synaptic modulator of this last neuron increases from 0.15 to 0.2 V (blue line Fig. 2b) and, similar to the dopaminergic neuron, the spiking threshold is overcome (blue line Fig. 2b). On the other hand, when there is no DA present in the system (and thus the first neuron is silent), the second serotonergic neuron remains silent too even when 5-HT is present (black line Fig. 2b). These mechanisms emulate the biological scenario where a DA neuron inhibits the activity of the 5-HT synapse. The experiment shows that the conductance modulation of the second synapse, mediated by 5-HT, is strictly controlled by the modulation of the DA-mediated synapse and so by the spiking behaviour of its neuron. Hence, the biology inspired transition from signal inhibition to transmission (non-spiking to spiking) of interdependent dopamine and serotonin interneurons is fully regulated by the presence of neurotransmitters. Using our modular neuromorphic system, we thus demonstrate a neurotransmitter-mediated pathway primitive for chemically activated and modulated spiking neural networks.

## Retina inspired sensory coding and neuromodulation

When considering the retina as a model for a biological system, light and neurotransmitters represent the external and internal stimuli that not only (Fig. 2) trigger and control signal transmission along a (cascading) neural pathway (activating specific brain regions), but also modulate the spike frequency of persistently active neurons. In the central nervous, the modulation of active neurons represents the biological mechanisms of external signal processing involving a cascade interaction of sensory coding and spike-rate encoding that enables learning and memory. To demonstrate this concept, we further combine both neuronal functionalities (sensory coding and neuromodulation) in our organic neuromorphic pathway, demonstrating

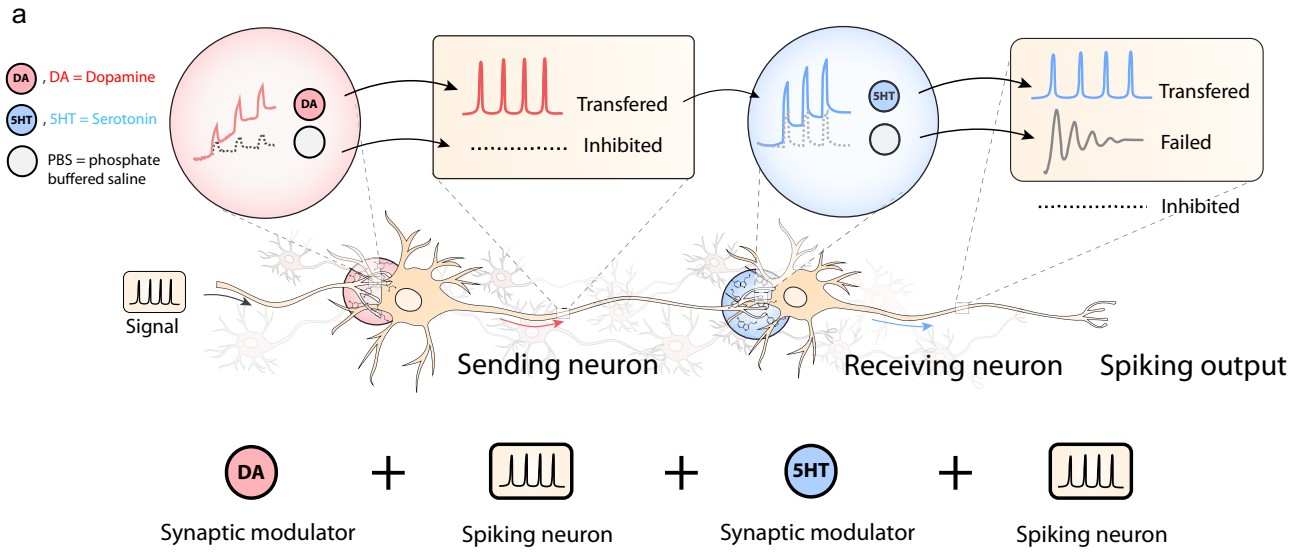

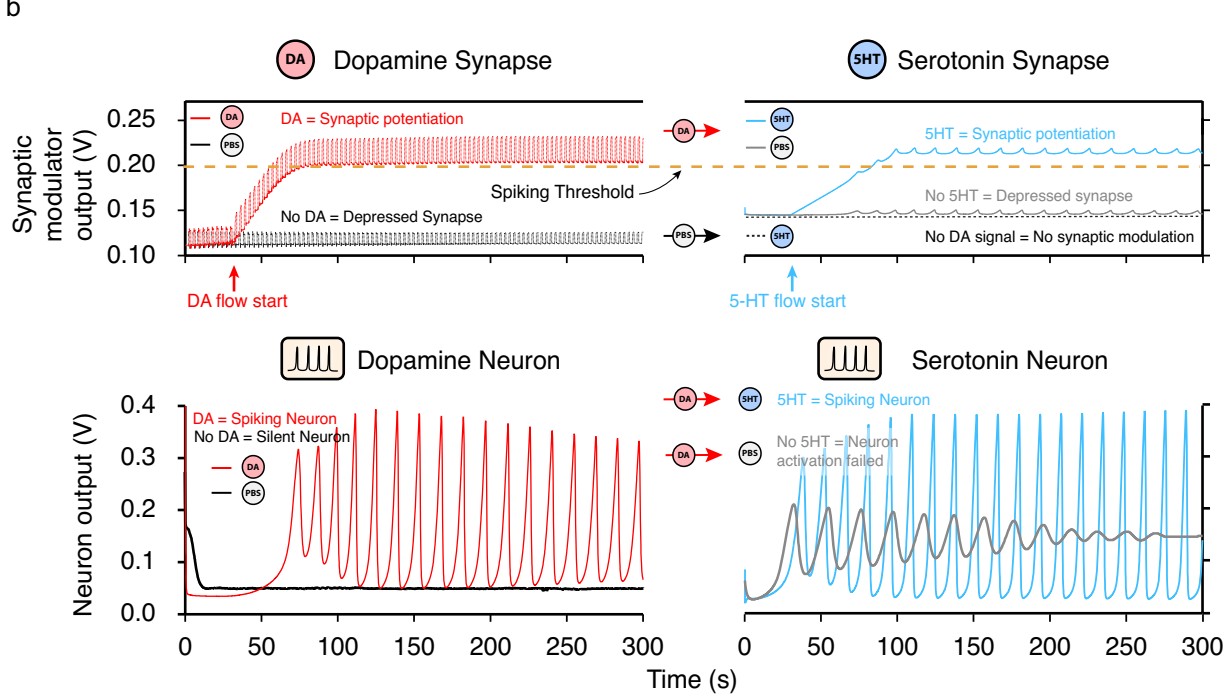

**Fig. 2 | Neural pathway of two coupled synapses. a** Illustration of the neural pathway between the sending, the dopaminergic and serotonergic neurons, highlighting the cases of signal transmission, inhibition, and failed activation. **b** Synaptic modulator output of both dopamine and serotonin neurons and spiking output of both neurons.

an intimate interaction between light spike-coding and neurotransmitter-mediated spike frequency modulations. As shown in Fig. 1c, the light intensity can be encoded in the neuron spike frequency so it can be sent to higher order computing units (e.g., through the optic nerve to the central nervous system). The power of the retina resides in its in-sensor functions, performed by the close-interaction of bipolar cells and amacrine cells. This allows the signal reaching the inner retina to be highly pre-processed, reducing the power consumption and increasing computing performance[47]. Some crucial steps in the pre-processing of visual information such as shape recognition and light adjustment mechanisms, only occur through the local action of neuromodulators, including dopamine[9]. The synthesis and activity of this neurotransmitter in dopaminergic amacrine cells (acting as interneurons), is controlled by the level of retina

illumination[48]. Hence, this interneuron modulates the signal coming from the afferent neurons. This signal can then be transmitted further to interneurons that are mediated by other neurotransmitters such as serotonin. To mimic this biological circuitry in our artificial retina prototype, we connected multiple neurons (spiking circuits) and their synapses (synaptic modulators) in cascade (Fig. 3a). The light-induced spike pattern of an afferent neuron is sent to the DA regulated synapse that modulates the synaptic plasticity and controls the activity of its neuron. Thereafter the signal travels further to the next serotonergic neuron, connected through its synapse, whose spike frequency is modulated by the neurotransmitter serotonin.

In Fig. 3b, we expose the light sensor to two light conditions (low and high intensity) that each result in a specific receptor potential that triggers a spike train with a certain frequency (Fig. 1c, see also

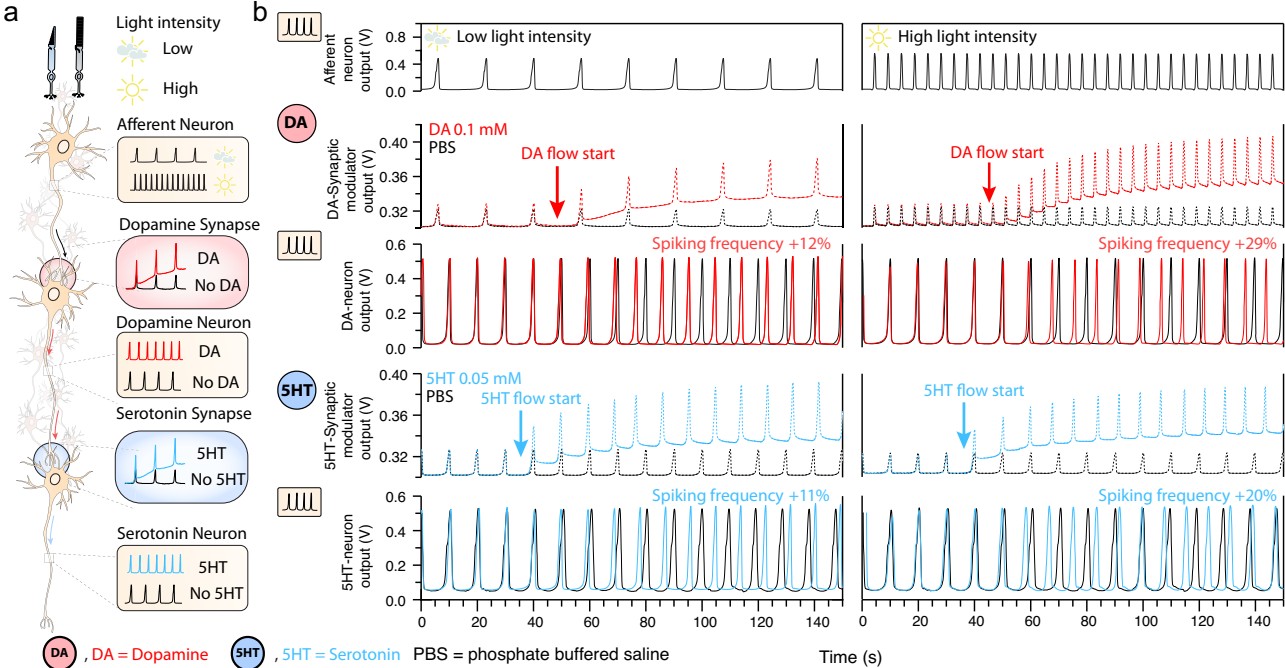

**Fig. 3 | Combining sensory coding and neuromodulation in a neuromorphic artificial retina. a** Schematics depicting the biological neural pathway. **b** Output voltage of the afferent neuron, the synaptic modulator of the dopamine neuron, the dopamine neuron, the synaptic modulator of the serotonin neuron, the serotonin neuron, for two light intensity conditions (low and high).

Supplementary Discussion 5). We use both these spike patterns as an input to the first dopaminergic neuron (see Fig. 3b), inducing a permanent modulation of the chemical synapse through the neuromodulation mechanism (Fig. 1d) in the case that the neurotransmitter was present (see Fig. 3b). These different spikes patterns affect the maximum synaptic modulator output variations (Supplementary Discussion 5). In the condition with high light intensity, the expression of DA at the first synapse leads to a 0.08 mV synaptic modulation (Fig. 3b), which results in a 29% increase of the spike frequency of the DA neuron (Fig. 3b). On the other hand, the low light intensity condition causes a 0.04 mV synaptic modulation and a corresponding spike frequency increase of 12%. Note that, similar to Fig. 1d, the output of the synaptic modulator shows spikes resulting from its input signal but only the baseline voltage determines the spike frequency of the output signal of the corresponding neuron. The signal from the spiking neuron when DA is present, is transmitted to the next serotonergic neuron. Here, the release of 5-HT causes a synaptic modulation of 0.05 mV (Fig. 3b) corresponding to a spike frequency increase of 20% (Fig. 3b) for a high light intensity while the low ambient light intensity leads to a 5-HT-mediated modulation of 0.035 mV and a spike frequency increase of 11%.

This neuromorphic cascading pathway enables the transduction of ambient light intensities to a voltage potential and subsequently to a precise spike pattern which translates to specific neurotransmitter-mediated synaptic modulations. As such, the biohybrid synapse of the DA synaptic modulator stores the information, received in the form of spikes, as a synaptic weight change (conductance modulation) and generates a corresponding spike pattern that elicits a cascade mechanism on the 5-HT neuron: a key to replicate the biological scotopic regulation mediated by retinal neurotransmitters.

More sophisticated, optical sensors including light wavelength selectivity[49] and multiplexing capabilities[50] can be implemented to further improve the performance of our system by integrating additional biology-inspired functions such as light-wavelength sensitivity (colour -photopic vision of cones) and the summation of stimuli from sensors working at different light intensities (rods and cones mesopic vision), respectively.

## Discussion

The modular chemically-adaptive neuromorphic system presented in this work demonstrates an integrated approach to build tailored biohybrid pre-processing units for programmable and cascadable neural pathways. We illustrate a strategy which leverages the Hebbian learning functions of neuromorphic devices to develop a modular and reconfigurable circuit displaying synaptic conditioning based on biochemical signalling activity (dopamine and serotonin) and light stimuli, by integrating light sensors, spiking circuits and synaptic modulators. This modularity allows for straightforward design of systems that can selectively trigger spike-encoded signal transmission, while locally transducing and computing both physical as well as physiological environment information: an essential step towards realizing complex neuromorphic hardware at the biohybrid interface. As such, this neuromorphic platform replicates the interrelated biological functions that are not only exploited to demonstrate loose and decoupled biomimetic concepts but to establish a primitive version of an artificial neuromorphic computing circuit.

We highlight that the modular elements enable an unprecedent interdependency of biological functions (sensory coding and neuromodulation) thus resembling the performance of biological pathways. Our circuit represents a prototype for more complex systems that can be designed to introduce Hebbian learning functions (based on the neuromodulatory activity of the biohybrid synapses), and, depending on frequency-selective rules, multi-sensory stimuli processing and integration based on the afferent neurons functions. We envision integrating algorithms for retina-specific computational functions to realise adaptive biohybrid computing pathways.

These neuromorphic pathways aim to replace damaged biological first order processing units (such as the one constituted by photoreceptors, bipolar and amacrine cells), thereby restoring the signal transmission to higher order units within the central nervous system.

## Methods

### Integrated neuromorphic circuit fabrication

Ti (5 nm)/Au(100 nm) electrodes were patterned on glass slide substrates by a bottom down approach. First the metals were deposited on glass via thermal evaporation, then the surface was cleaned through plasma cleaning and covered with the positive photoresist S1805. The electrode patterning was performed with the use of the Maskless Aligner - Heidelberg MLA150. After development the metals were etched via chemical etching using first gold etchant (30 s) and then HCl (5 h) for removing Ti. After a surface treatment with the adhesion promoter Silane A-174, a 2.3 μm -thick parylene C was deposited on the substrate serving as an encapsulation layer. Then, a diluted micro-90 (2% v/v in DI water) was spin-coated as an anti-adhesive layer, and subsequently, a sacrificial second parylene layer of 2.3 μm was deposited. The electrode areas were opened through photolithography (SU-8 3005). The overall circuit design is illustrated in Supplementary Discussion 1, while the fabrication steps are depicted in Supplementary Discussion Fig. 2. The electrodes were covered by spin coating with the different materials avoiding cross-contamination by placing on the substrate a PDMS well to confine the desired solutions.

The interdigitated microelectrodes used for the four OECTs of the inverter pair and one for the switch have an electrode width of 10 m and distance 5 m, and comprises 22 finger pairs with length 2.2 mm. The circuit design includes also a capacitor which was disconnected during the measurements since it did not contribute to the response time of the circuit, as the capacitances of the other elements (switch and inverter OECTs) were dominant. The capacitor is constituted by two areas with dimension 250 × 500 m where PEDOT:PSS solution was spin-coated. See Supplementary Discussion 1 for a schematic description of the fabrications steps.

### p(C$_4$-T2-C$_0$-EG) devices spin coating

The polymer p(C$_4$-T2-C$_0$-EG) was synthesised by McCulloch group (see Supplementary Discussion 1)[51]. The material solutions were prepared in chloroform at the concentration of 20 mg mL$^{-1}$. The interdigitated microelectrodes, and so the whole circuit substrate, were pre-treated with UV ozone over 15 min. The polymer solutions were spin-coated at 500 rpm for 60 s to fabricate the OECT comprising the inverters of the spiking circuit.

### P-3O devices fabrication

The polymer P-3O was synthesised by Gang Ye following a previously reported protocol and the material solutions were prepared in chloroform at the concentration of 5 mg mL$^{-1}$ [35]. The interdigitated microelectrodes were pre-treated with UV ozone over 15 min. The polymer solutions were spin-coated at 1000 rpm for 30 s.

### Biohybrid synapse

The biohybrid synapses were fabricated on glass-ITO patterned substrates purchased from Xin Yan Technology Ltd. The size of the glass square substrates is 25 mm with patterned squares of ITO (20 ohm sq$^{-1}$) of 10 mm covering each corner. Substrates were cleaned through sonication in IPA (Sigma-Aldrich, USA) for 20 min. Employing Kapton tape and using a PMMA master mask, 2 stripes connecting opposed ITO squares (with fixed width 5 mm) were traced for the deposition of the polymer mixture. PEDOT:PSS (Hereaus, Clevios PH 1000) aqueous solution was prepared by adding 6 vol.% ethylene glycol (Sigma-Aldrich, USA) to increase the PEDOT:PSS conductivity, 0.1 vol.% dodecylbenzene sulfonic acid (Sigma-Aldrich, USA) as a surfactant, and 1 vol.% (3-glycidyloxypropyl)trimethoxysilane (Sigma-Aldrich, USA) as a crosslinking agent to improve mechanical stability. PEDOT:PSS solution was spun on the selected areas of the substrate at 1000 rpm for 2 min and baked at 120 °C for 20 min. Before operation the devices were conditioned at least 20 min in PBS solution, in order to avoid swelling effects during electrical measurements.

### Neurotransmitters solutions preparation

Neurotransmitter solutions were freshly prepared by dissolving dopamine hydrochloride (98%, Sigma-Aldrich, USA) and serotonin hydrochloride (98%, Sigma-Aldrich, USA) into Dulbecco's Phosphate Buffered Saline (Modified, without calcium chloride and magnesium chloride, Sigma-Aldrich, USA) at multiple concentrations ranging from 0.01 mM to 0.1 mM.

### Connected Spiking circuits and electrical measurements

Each integrated spiking circuit was connected to AKEO multichannel using pogo-pin clamps as depicted in Supplementary Discussion 4 and 5. For each circuit $V_{DD1}$ and $V_{DD2}$ and $V_{IN}$ were applied through ARKEO channels while simultaneously recording the $V_{OUT}$. For the experiment in Fig. 2, the trace of the afferent neuron $V_{OUT}$ was recorded in a preliminary measurement. After connecting two neurons operating with DA and 5-HT, the recorded trace was used as $V_{IN}$ to the hybrid synapse of the DA-neuron using ARKEO multichannel as a voltage source. All the different configuration of the neurons connections and schematic, including the microfluidic setup are described in Supporting Discussion 4 and 5.

## Data availability

The data represented in Figs. 1, 2 and 3 are provided with the paper as source data. Additionally, the data represented in the Supplementary Information Figs are provided in the same format. Other datasets generated and/or analysed during the current study are available from the authors on request. Source Data are provided with this paper.

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

## Acknowledgements

E.R.W.D. and Y.v.d.B. acknowledge financial support from The European Union's Horizon 2020 Research and Innovation Programme, grant agreement no. 802615. This research was funded in part, by the European Union's Horizon 2020 research and innovation programme under grant agreement n°952911, project BOOSTER, grant agreement n°862474, project RoLA-FLEX and grant agreement n°101007084 CITYSOLAR, as well as EPSRC Project EP/T026219/1 EP/W017091/1. We gratefully acknowledge Alberto Salleo as a source of inspiration for the spiking circuit approach. The authors thank Bastiaan de Jong and Lucio Cinà of Cicci Research who helped with the design and implementation of the custom post processing code for the Arkeo platform.

## Author contributions

G.M.M. and Y.v.d.B. conceptualized the research and established the theoretical approach. F.S. contributed to the bio-hybrid synapses design and data analysis. G.M.M. and E.R.W.v.D. designed and performed the experiments while they both also analysed the data and prepared the manuscript. I.M., S.G. and G.Y. synthesised the materials as reported in Methods. J.R. and A.S. designed the monolithically integrated spiking circuit and assisted during the device fabrication. Z.L. developed the LT spice model included in the Supplementary Information.

## Competing interests

The authors declare no competing interests.
