## [Peer Review File · Nature Communications]

A modular organic neuromorphic spiking circuit for retina-inspired sensory coding and neurotransmitter-mediated neural pathwaysEditorial Note: This manuscript has been previously reviewed at another journal that is not operating a transparent peer review scheme. This document only contains reviewer comments and rebuttal letters for versions considered at *Nature Communications*.

REVIEWER COMMENTS

Reviewer #1 (Remarks to the Author):

The authors addressed most of the concerns. One remaining is to make a clear comparison between organic and inorganic materials, or between organic non-electrolyte and electrolyte materials, to fully prove the benefits of the current materials selection regarding interneurons mimicking.

Reviewer #2 (Remarks to the Author):

In the transferred manuscript, Giovanni et al. has made further revisions according to the reviewers' comments. Several remaining comments still need to be addressed before this work can be considered for publication in *Nature Communications*.

1) Demonstrating the concept of a fully organic neural pathway is an important task, but this work seems to be simply cascading organic synapses and organic neuronal circuits together. Beyond the prototype system-level demonstration, the scientific innovations and new phenomena discovered to realize such neural pathway are more important and critical, which should be emphasized and discussed in details. It is suggested that the authors further clarify the key innovations of this work. In the present form, building a multi-device cascading neuromorphic system with superficial connection to biomimetic concepts seems insufficient.

2) In addition, the advantage of neural and synaptic structures based on organic materials lies in their biocompatibility, so the prospects of their biological applications need to be carefully considered. Otherwise, it will significantly reduce the significance of this work. Therefore, to what extent it resembles the biological system (such as frequency) as mentioned by the previous reviewers is all based on this consideration. This does not necessarily require the neuromorphic system to surpass the performance of biological systems, but rather to clarify the compatibility between current technological levels and key performance metrics. Therefore, a quantitative comparison between the proposed neuromorphic pathway and the corresponding biological counterpart is still highly recommended. Moreover, the adjustable levels of DA and 5HT in the human body should also be listed as a reference for the proposed biological system.

Replies to Reviewers comments.

Reviewer 1

1. The authors addressed most of the concerns. One remaining is to make a clear comparison between organic and inorganic materials, or between organic non-electrolyte and electrolyte materials, to fully prove the benefits of the current materials selection regarding interneurons mimicking.

We thank the reviewer for the comment, and we acknowledge the need for a clarification of the part of the manuscript illustrating the benefits of organic materials to the design of the presented neuromorphic system while providing a comparison to inorganic and organic non-electrolyte materials-based platforms. We have therefore expanded and modified the introduction section, investigating the limits of organic and inorganic neuromorphic devices that we believe to address with our work. We here report the afore-mentioned section:

“Although organic non-electrolyte conductors have been previously employed to build spiking circuits based on organic field effect transistors (OFETs), their lack of ionic conduction (interaction with aqueous electrolytes) severely limit their applicability towards adaptive neuromorphic circuits¹. Standard OFETs operate in dry conditions and do not require an electrolyte medium, which in the context of neuromorphic and smart bioelectronic applications represents the bio-interfaced environment interacting with the platform². Moreover, to act as biosensors these devices usually require surface functionalisation and suffer from limitations related to low sensitivity and selectivity.

Mixed ionic-electronic conductors allow to build organic electrochemical transistors (OECTs) where ion injection from an electrolyte modulates the bulk conductivity the channel terminal. This mechanism allows to design neuronal circuits that closely match the operating timescales of their biological counterparts³, reduce interface impedance⁴, and mimic ion-based biological functions such as neuronal ion-flux communication and neurotransmitter-receptor binding⁵ (synaptic function), which are required to interact with biological tissues and to design adaptive biointerfaces⁶. Recently, organic materials have been employed to build electronic circuits that mimic the spiking behaviour of neurons^{1,7,8}. In this context, mixed ionic-electronic conductors-based neuromorphic devices are emerging due to their intrinsic ion tunability which elicit a change of their electrical characteristics⁹. The distinctive interdependence of these conduction mechanisms (ionic to electronic) is essential to replicate the chemical (neurotransmitters) to electrical (action potential) signal transduction processes “in-sensor” and is performed by the two key biological elements considered in this work: synapses and neurons^{8,10,11}. On the contrary, to perform similar biological functions inorganic and OFETs-based circuits devices require a large number of circuitual elements with limited tissue adaptation and complex external sensing units. These require multiple signal transmission optimisation mechanisms (amplification, reduction, noise control), significantly limiting the reconfigurability of the overall circuit.^{12-14.”}

Moreover, we have also introduced in the Supplementary Discussion 1 a table (Table 1) which anticipates some of the metrics reported in Table 14 to highlight the comparison between inorganic materials (grey rows), organic non-electrolyte materials (OFET-based, green rows), and organic electrolyte materials (OECT-based, brown rows) neuromorphic systems. Accordingly, the introduction of Supplementary Discussion 1 now focus on these materials comparison and we here report the amended version:

“This section introduces the devices and materials that compose the spiking circuits (Supplementary Figure 1 a,b). Before presenting the single devices characteristics, an overview is provided describing the state-of-the-art materials that are commonly used to design neuromorphic spiking circuits, highlighting the benefits of employing organic electrolyte devices/materials (OECT) comparing organic non-electrolyte (OFET) and inorganic technologies (Table 1).

The circuitual design of current spiking circuits is predominantly based on inorganic materials, particularly on silicon-based devices¹⁵ (Supplementary Discussion 1, Table 1). However,

neuromorphic applications operating at the biointerface clearly favour organic materials⁵, and specifically OECT-based neuromorphic circuits. Next to their relatively soft and flexible properties, high tunability and low operational voltage, organic mixed ion-electron conductors conduct both electrons and ions¹⁶. This mixed conduction allows organic mixed conducting materials to closely match the operating timescales of their biological counterparts³, reduce interface impedance⁴, and mimic ion-based biological functions such as neuronal ion-flux communication and neurotransmitter-receptor binding⁵ (synaptic function), which are required to interact with biological tissues and to design adaptive biointerfaces⁶. Although showing desirable biocompatibility and flexibility properties, standard OFETs materials fail to “translate” (with the same sensibility and reliability of OECTs) the interaction of a complex bio-interfaced environment into a change of the neuromorphic circuit electrical characteristics.”

Table 1 List of the critical characteristics of artificial neurons and neuromorphic circuits based on different technologies, comparing inorganic materials (grey), organic non-electrolyte materials (OFET-based, green), and organic electrolyte materials (OECT-based, brown) neuromorphic systems.

	Circuit elements	Features	Footprint μm^2	Switching speed (single device)	Spike voltage amplitude [mV]	Operative Frequency [Hz]	Power or Energy consumption per spike	Operation in ionic environment	Connectivity to additional neurons
Si CMOS (based on integrate and fire or ¹⁷⁻²¹)	From 10 to 30+ transistors ¹⁹	~ 20	993 μm^2	< 1ns	120 ²⁰ to 2300 ¹⁹	30 ²¹	< 100 fJ to 1nJ	no	yes
Mott-memristors ²²⁻²⁶	2 memristors 1 capacitor 2 resistors	~ 20	100 μm^2 ²³	< 1ns	300 ²⁴ - 3300 ²²	10 to 10 kHz	~ 100 nW	no	yes
2D material Gaussian junction ²⁷ and heterojunction ^{28,29}	>10transistors 3 resistors 3 capacitors	8 simulation	0.25 cm^2	-	450	< 1	250 nJ	no	no
OFET LIF ^{1,30,31}	3 resistors 3 transistors 2 capacitors	3	~ 10 mm^2	-	3000	200	20 μJ	no	yes
Complementary LIF ^{8,32}	5 transistors 1 capacitor	3	~ 10 mm^2	15 ms	4000	~ 2	0.5-40 nW	yes	no
Complementary LIF (this work)	5 transistors	3	~ 1 mm^2	120 ms	4000	~ 0.25	0.1 μJ	yes	yes

Additionally, we now anticipate in the abstract the relevance of organic mixed ionic-electronic conductors for the design of the presented neuromorphic platform. Hence, we have introduced in the abstract a clear reference to the benefits of this materials which are demonstrated through this work:

“Here, we leverage the intrinsically interrelated charge transport properties of mixed ionic-electronic conductors to reproduce the chemical-to-electrical signal transduction processes of biological circuits”.

Reviewer 2

1. Demonstrating the concept of a fully organic neural pathway is an important task, but this work seems to be simply cascading organic synapses and organic neuronal circuits together. Beyond the prototype system-level demonstration, the scientific innovations and new phenomena discovered to realize such neural pathway are more important and critical, which should be emphasized and discussed in details. It is suggested that the authors further clarify the key innovations of this work. In the present form, building a multi-device cascading neuromorphic system with superficial connection to biomimetic concepts seems insufficient.

We are grateful to the reviewer for the insightful comment that gave us the opportunity to improve the manuscript. We believe the new phenomena discovered and leveraged in this manuscript (as highlighted by the comment) allow for the first time to build a reconfigurable neuromorphic circuit which has the potential to replicate different neuronal pathways. The

cascade connection of multiple spiking circuits and synapses proves that the reconfigurable and modular approach used to build our circuits can be extended to realise more complex and extended neuromorphic platforms, ideally approaching (facing additional technological challenges) a bio-inspired network.

However, we believe the core achievement of this manuscript is the design of spiking units and synapses that can be used to replicate “interdependent connections” among different neurons in the human brain. The change in spike patterns of different neuronal units replicate both sensory coding (from light stimuli) and interneuron modulation (depending on dopamine and serotonin concentrations), and it is achieved by translating environmental and chemical information into spike patterns, respectively. These two phenomena replicate two core biological functions that allows to move from mere sensing of external/internal environment conditions to a bio-inspired intrinsic combination of memory and computation of these two inputs which is clearly based on a spike-based code. Moreover, in the manuscript section “**Interdependent neural pathway between dopaminergic and serotonergic neurons**” we specifically replicate the interdependent neural connection between two interneurons, where a spiking signal is transferred or inhibited by the sending neuron whose spiking activity, in turn, triggers the spiking of the next interneuron (receiving neuron) in the biological circuit. This experiment was inspired by biology as DA neurons in the ventral tegmental area bidirectionally regulate the activity of 5-HT neurons in the dorsal raphe nucleus³³. This demonstrate the potential of our modular systems, beyond the cascading of spike patterns modulations, to design more complex and interconnected networks which are able to selectively initiate signals and so perform pre-processing functions.

Considering the above, we have amended the **Discussion** section of the manuscript to illustrate this work achievements and particularly the in-sensor computing potential of our neuromorphic platform beyond the superficial connection to loose biomimetic concepts. We report the new version of this section:

“The modular chemically-adaptive neuromorphic system presented in this work demonstrates an integrated approach to build tailored biohybrid pre-processing units for programmable and cascable neural pathways. We illustrate a strategy which leverages the Hebbian learning functions of neuromorphic devices to develop a modular and reconfigurable circuit displaying synaptic conditioning based on biochemical signaling activity (dopamine and serotonin) and light stimuli, by integrating light sensors, spiking circuits and synaptic modulators. This modularity allows for straightforward design of systems that can selectively trigger spike-encoded signal transmission, while locally transducing and computing both physical as well as physiological environment information: an essential step towards realizing complex neuromorphic hardware at the biohybrid interface. As such, this neuromorphic platform replicates for the first time the interrelated biological functions that are not only exploited to demonstrate loose and decoupled biomimetic concepts but to establish a primitive version of an artificial neuromorphic computing circuit.

We highlight that the modular elements enable an unprecedented interdependency of biological functions (sensory coding and neuromodulation) thus resembling the performance of biological pathways. Our circuit represents a prototype for more complex systems that can be designed to introduce i) Hebbian learning functions based on the neuromodulatory activity of the biohybrid synapses and depending on frequency-selective rules ii) multi-sensory stimuli processing and integration based on the afferent neurons functions. We envision integrating novel algorithms for retina-specific computational functions to realise adaptive biohybrid computing pathways.

These neuromorphic pathways aim to replace damaged biological first order processing units (such as the one constituted by photoreceptors, bipolar and amacrine cells), thereby restoring the signal transmission to higher order units within the central nervous system.”

2. In addition, the advantage of neural and synaptic structures based on organic materials lies in their biocompatibility, so the prospects of their biological applications need to be carefully considered. Otherwise, it will significantly reduce the significance of this work. Therefore, to what extent it resembles the biological system (such as frequency) as mentioned by the previous reviewers is all based on this consideration. This does not necessarily require the neuromorphic system to surpass the performance of biological systems, but rather to clarify

the compatibility between current technological levels and key performance metrics. Therefore, a quantitative comparison between the proposed neuromorphic pathway and the corresponding biological counterpart is still highly recommended. Moreover, the adjustable levels of DA and 5HT in the human body should also be listed as a reference for the proposed biological system.

We are grateful to the reviewer for the comment, and we acknowledge the need for a clarification about the the prospects of future potential biological applications of our neuromorphic reconfigurable platform.

As from Supplementary Discussion 3, the neuromorphic system demonstrates a reconfigurable network of neurons which can be connected through chemical synapses (biohybrid synapses). These elements are key to adjust the synaptic weights which is represented by the conductance state of the OECT of the synaptic modulator. The synaptic weight is updated depending on the neurotransmitter environment and serves to directly regulate the spike frequency of a connected (in series) spiking circuit. These synapse-to-neuron (artificial synapse to spiking circuit) connections can be configured to design cascaded networks which resemble neural pathways.

The significance of this work, as stated by the reviewer, resides in the possibility not only to replicate biological functions through a neuromorphic pre-processing unit but more importantly to establish an intimate and adaptive interaction between this former and biological circuits.

On this note, the reconfigurable network allows “by design” future bio-hybrid scenarios (cells to device coupling).

To address potential biomimicry concerns, we have expanded Supplementary Discussion 5 introducing new concepts and arguments which are also discussed in Supplementary Discussion 3 (neurotransmitter-mediated plasticity) and Supplementary Discussion 7 (spikes frequency variations):

“We here contextualize the platform’s performance and functions with emphasis on the presented cascaded neural pathway, establishing a direct parallel to specific biological circuits displaying similar functions and thus commenting on its applicability as a bio-interfaced system. Building on Supplementary Discussion 3, we still consider the work by Keene et al. demonstrating the use of biohybrid synapses as devices allowing the transduction of chemical signals from PC12 cells (released dopamine) into permanent conductance modulations of the organic neuromorphic device’s channel terminal. This work established a chemical to electrical signal transduction approach which has been thoroughly investigated from an electrochemical perspective by Matrone et al.³⁴. In our synaptic modulator, dopamine and serotonin allow a semi-permanent conductance modulation (synaptic weight update) which is used to control the activity of a connected spiking neuron, eliciting not only frequency modulations in persistently spiking neural networks (Fig.1 and 3) but also triggering spikes (neural activation), and so signal transmission, in a silent pathway (Fig.2). By design, the biohybrid synapse used in the synaptic modulator can be employed as an active interface with tissue and cells models. Indeed, leveraging the biocompatibility of PEDOT:PSS, cells can be directly seeded on the gate terminal of this device, with extensive reports supporting these applications in literature^{35–37}. In this configuration, the presence of physiologically released electroactive neurotransmitters, such as dopamine and serotonin, allows to operate the presented neuromorphic neural pathway as an adaptive platform which does not merely record biochemical clues into electrical signals but more importantly locally transduce these relevant signals in a “neuromorphic fashion”.

By examining biological neural pathways involving a complex interconnection of neuromodulation processes, we establish a direct parallel between the neuromorphic platform and midbrain neurons thus validating the significance of this work.

DA neurons in the ventral tegmental area bidirectionally regulate the activity of 5-HT neurons in the dorsal raphe nucleus, a biological scenario which has been replicated through the neuromorphic platform and is illustrated in Fig.2 and Fig.3.³³ We investigated the biological phenomena at the base of the interaction between dopamine and serotonin neurons, and hereby highlight technological gaps which must be addressed to establish an intimate coupling between neuromorphic devices and biological pathways. First, as analyzed in detail in Supplementary Discussion 7, we reaffirm that only the amplitude of change of the spikes

frequency represent the “brain encoding language” which allow to translate both external and internal stimuli into neural data. As such, midbrain interneurons perform computing functions through frequency modulations of +/- 30% (respect to the baseline frequency, Supplementary Fig. 30), which correspond to the same frequency changes demonstrated by our neuromorphic platform in Fig. 2 and 3. On the other hand, viable strategies to increase the operative frequency of the neuromorphic systems are investigated and suggested in Supplementary Discussion 7. Finally, focusing on the electrical to chemical signal transduction, we here investigate the concentrations of neurotransmitters involved in biological neurons functions, to prove the neuromorphic platform applicability in bio-hybrid scenarios.

For DA and 5-HT, two kind of signalling patterns (in the midbrain and retina) have been identified as phasic and tonic, supposedly associated to the biological functions of teaching signal (direct signal processing) and motivational drive (indirect signal processing), respectively. Direct signal processing is associated to the release of neurotransmitters in the pre- to post-synaptic space (4 nm) upon pre-synaptic firing (neurons phasic activity). This mechanism represents a core computational primitive which we intended to replicate in this work. According to the suggestion that the synaptic compartment has to be distinguished from the extra-synaptic compartment, synaptic transmission can be modeled on the hypotheses that (1) neurotransmitters release is highly localized to the synapses and that (2) neurotransmitters uptake strongly contributes to this tight localization³⁹. Under these assumptions, the concentration of neurotransmitters peaks during firing and returns to the baseline level (4-50 nM) in the ms range. While it is still difficult to estimate or directly measure the peak concentration reached during a firing event, models predict a change of the instantaneous (local) level of neurotransmitters to be in the tens of mM. In this work we used concentrations of neurotransmitters ranging from 0.025 mM to 0.1 mM (25 to 100 μ M) to simulate the phasic release of dopamine and serotonin at the synaptic terminal. While lower concentrations of neurotransmitters (below 10 μ M) can be still used to elicit the conductance modulation phenomena described in Supplementary Discussion 3, the range of concentrations employed in this work still correspond to a realistic biological scenario which can be replicated by bio-interfacing the platform with neuronal model cells such as PC12 (dopamine release). As such, midbrain and retina neuronal pathways depending on the interaction of afferent and interneurons mediated by the neurotransmitter dopamine and serotonin have been replicated by the design of the modular neuromorphic platform.”

Accordingly, we have introduced a direct reference in the main text to the former section, introducing the biological to artificial pathway parallel motivations:

“When an electro-active neurotransmitter, such as dopamine or serotonin, is present in the electrolyte, the oxidation of these species leads to a non-volatile conductance modulation (synaptic weight modulation) ^{10,34}. This mechanism and the use of a fluidic module (Supplementary Discussion 3) replicate the phasic release of neurotransmitters (in the midbrain and retina), which is assumed to be associated to the biological functions of signal processing leading to learning and memory.³⁹ Signal computing at the synaptic level depends on the fluctuation of neurotransmitters concentration in the mM range at the synaptic cleft (the pre- to post-synaptic 4 μ m space), as these are released upon pre-synaptic firing (Supplementary Discussion 3 and 5).”

1. Mirshojaeian Hosseini, M. J. *et al.* Organic electronics Axon-Hillock neuromorphic circuit: towards biologically compatible, and physically flexible, integrate-and-fire spiking neural networks. *J. Phys. D: Appl. Phys.* **54**, 104004 (2021).
2. Niu, Y. *et al.* Expanding the potential of biosensors: a review on organic field effect transistor (OFET) and organic electrochemical transistor (OECT) biosensors. *Mater. Futures* **2**, 042401 (2023).

3. Simon, D. T., Gabrielsson, E. O., Tybrandt, K. & Berggren, M. Organic Bioelectronics: Bridging the Signaling Gap between Biology and Technology. *Chem. Rev.* **116**, 13009–13041 (2016).
4. Owens, R. M. & Malliaras, G. G. Organic Electronics at the Interface with Biology. *MRS Bulletin* **35**, 449–456 (2010).
5. Gumyusenge, A., Melianas, A., Keene, S. T. & Salleo, A. Materials Strategies for Organic Neuromorphic Devices. *Annu. Rev. Mater. Res.* **51**, 47–71 (2021).
6. Bettucci, O., Matrone, G. M. & Santoro, F. Conductive Polymer-Based Bioelectronic Platforms toward Sustainable and Biointegrated Devices: A Journey from Skin to Brain across Human Body Interfaces. *Adv. Mater. Technol.* 2100293 (2021) doi:10.1002/admt.202100293.
7. Harikesh, P. C. *et al.* Ion-tunable antiambipolarity in mixed ion–electron conducting polymers enables biorealistic organic electrochemical neurons. *Nat. Mater.* **22**, 242–248 (2023).
8. Harikesh, P. C. *et al.* Organic electrochemical neurons and synapses with ion mediated spiking. *Nat Commun* **13**, 901 (2022).
9. Gkoupidenis, P. *et al.* Organic mixed conductors for bioinspired electronics. *Nat Rev Mater* (2023) doi:10.1038/s41578-023-00622-5.
10. Keene, S. T. A biohybrid synapse with neurotransmitter-mediated plasticity. *Nature Materials* **19**, 16 (2020).
11. Sarkar, T. *et al.* An organic artificial spiking neuron for in situ neuromorphic sensing and biointerfacing. *Nat Electron* **5**, 774–783 (2022).
12. De Donno, D., Catarinucci, L. & Tarricone, L. RAMSES: RFID Augmented Module for Smart Environmental Sensing. *IEEE Trans. Instrum. Meas.* **63**, 1701–1708 (2014).
13. Andersson Ersman, P. *et al.* All-printed large-scale integrated circuits based on organic electrochemical transistors. *Nat Commun* **10**, 5053 (2019).
14. Pecqueur, S. *et al.* Neuromorphic Time-Dependent Pattern Classification with Organic Electrochemical Transistor Arrays. *Adv. Electron. Mater.* **4**, 1800166 (2018).
15. Serb, A. *et al.* Memristive synapses connect brain and silicon spiking neurons. *Scientific Reports* **10**, 1–7 (2020).
16. Paulsen, B. D., Tybrandt, K., Stavrinidou, E. & Rivnay, J. Organic mixed ionic–electronic conductors. *Nature Materials* **19**, 13–26 (2020).

17. Akbari, M., Hussein, S. M., Chou, T.-I. & Tang, K.-T. A 0.3-V Conductance-Based Silicon Neuron in 0.18 μm CMOS Process. *IEEE Trans. Circuits Syst. II* **68**, 3209–3213 (2021).
18. Livi, P. & Indiveri, G. A current-mode conductance-based silicon neuron for address-event neuromorphic systems. in *2009 IEEE International Symposium on Circuits and Systems* 2898–2901 (IEEE, 2009). doi:10.1109/ISCAS.2009.5118408.
19. Wijekoon, J. H. B. & Dudek, P. Compact silicon neuron circuit with spiking and bursting behaviour. *Neural Networks* **21**, 524–534 (2008).
20. Rubino, A., Payvand, M. & Indiveri, G. Ultra-Low Power Silicon Neuron Circuit for Extreme-Edge Neuromorphic Intelligence.
21. Vuppunuthala, S. & Pasupureddi, V. S. 3.6-pJ/Spike, 30-Hz Silicon Neuron Circuit in 0.5-V, 65 nm CMOS for Spiking Neural Networks. *IEEE Trans. Circuits Syst. II* 1–1 (2023) doi:10.1109/TCSII.2023.3324584.
22. Pickett, M. D. A scalable neuristor built with Mott memristors. *NATURE MATERIALS* **12**, (2013).
23. Yi, W. *et al.* Biological plausibility and stochasticity in scalable VO₂ active memristor neurons. *Nat Commun* **9**, 4661 (2018).
24. Li, X. *et al.* A Memristors-Based Dendritic Neuron for High-Efficiency Spatial-Temporal Information Processing. *Adv. Mater.* (2023).
25. Amin Fida, A., Khanday, F. A. & Mittal, S. An active memristor based rate-coded spiking neural network. *Neurocomputing* **533**, 61–71 (2023).
26. Zhao, J. *et al.* Memristors based on NdNiO₃ nanocrystals film as sensory neurons for neuromorphic computing. *Mater. Horiz.* **10**, 4521–4531 (2023).
27. Thakar, K., Rajendran, B. & Lodha, S. Ultra-low power neuromorphic obstacle detection using a two-dimensional materials-based subthreshold transistor. *npj 2D Mater Appl* **7**, 68 (2023).
28. Beck, M. E. *et al.* Spiking neurons from tunable Gaussian heterojunction transistors. *Nat Commun* **11**, 1565 (2020).
29. Yan, X. *et al.* Reconfigurable mixed-kernel heterojunction transistors for personalized support vector machine classification. *Nat Electron* (2023) doi:10.1038/s41928-023-01042-7.
30. Mirshojaeian Hosseini, M. J. *et al.* An organic synaptic circuit: toward flexible and biocompatible organic neuromorphic processing. *Neuromorph. Comput. Eng.* **2**, 034009 (2022).

31. Tischler, V. An integrate-and-fire neuron circuit made from printed organic field-effect transistors. *Organic Electronics* (2023).
32. Wu, H.-Y. *et al.* Stable organic electrochemical neurons based on p-type and n-type ladder polymers. *Mater. Horiz.* **10**, 4213–4223 (2023).
33. Cai, X. *et al.* A D2 to D1 shift in dopaminergic inputs to midbrain 5-HT neurons causes anorexia in mice. *Nat Neurosci* **25**, 646–658 (2022).
34. Matrone, G. M. *et al.* Electrical and Optical Modulation of a PEDOT:PSS-Based Electrochemical Transistor for Multiple Neurotransmitter-Mediated Artificial Synapses. *Adv Materials Technologies* 2201911 (2023) doi:10.1002/admt.202201911.
35. Broccard, F. D., Joshi, S., Wang, J. & Cauwenberghs, G. Neuromorphic neural interfaces: from neurophysiological inspiration to biohybrid coupling with nervous systems. *J Neural Eng* **14**, 041002 (2017).
36. Marzocchi, M. *et al.* Physical and Electrochemical Properties of PEDOT:PSS as a Tool for Controlling Cell Growth. *ACS Appl. Mater. Interfaces* **7**, 17993–18003 (2015).
37. Cellot, G. *et al.* PEDOT:PSS Interfaces Support the Development of Neuronal Synaptic Networks with Reduced Neuroglia Response In vitro. *Front Neurosci* **9**, 521 (2015).
38. Rathi, N. *et al.* Exploring Neuromorphic Computing Based on Spiking Neural Networks: Algorithms to Hardware. *ACM Comput. Surv.* **55**, 1–49 (2023).
39. Wiencke, K., Horstmann, A., Mathar, D., Villringer, A. & Neumann, J. Dopamine release, diffusion and uptake: A computational model for synaptic and volume transmission. *PLoS Comput Biol* **16**, e1008410 (2020).

REVIEWERS' COMMENTS

Reviewer #1 (Remarks to the Author):

I have no further comments.

Reviewer #2 (Remarks to the Author):

The authors have made further clarifications on the novelty of this work. I do not have any major comments. One minor issue is that, in the Supplementary Information, it appears Table 1 is just a selected list of Table 13, and the content is almost repeated. It is hence suggested to just combine them into one to avoid redundancy.

Response to Reviewers

Reviewer 2

We are grateful to the reviewer for pointing out a possible source of confusion. Hence, we have combined the two Tables (1 and 14) into the new Table 1 in Supplementary Discussion 1.